# GeoBIS: Budget-Optimal Block Importance Sampling for Stochastic Riemannian Optimization

## Abstract

This paper studies budgeted block subsampling for stochastic Riemannian optimization. Starting from the Horvitz–Thompson estimator, we derive an independent Bernoulli design with a water-filling probability rule that minimizes the second moment under a fixed expected number of active blocks. The resulting estimator, GeoBIS, is unbiased and achieves the canonical inverse-in-budget behavior of its second moment. We also analyze exact-$K$ negatively dependent designs, including projection determinantal point processes and sampling without replacement with unequal probabilities. Under a mild alignment condition on block directions, exact-$K$ strictly reduces the cross term in the variance. A simple wall-clock model provides a closed-form rule for selecting the active-block budget and clarifies when exact-$K$ is worthwhile. Experiments on orthogonality-constrained sequence models and thin-Stiefel adapters follow the predicted trends and validate GeoBIS as a practical default.

## 1 Introduction

Optimization with manifold constraints appears in many modern machine learning systems. Examples include orthogonal and Stiefel constraints in sequence models and adapters, subspace learning and matrix factorizations, and modules on product or hyperbolic manifolds. In these settings, the Riemannian gradient lives in a tangent space that admits natural block decompositions: coordinate groups, skew-rotation atoms such as Givens or Householder generators, or per-factor blocks in product manifolds. Executing full tangent updates can be expensive, and a common engineering strategy is to activate only a subset of blocks at each step and to reweight the contribution of sampled blocks using Horvitz–Thompson scaling so that the gradient estimator remains unbiased.

This paper addresses the design of such block subsampling policies when a fixed compute budget limits the expected number of active blocks per step. The first question is whether one can choose marginal probabilities so that the resulting estimator minimizes its second moment for a given budget. The second question is whether using a fixed-size subset with negative dependence can further reduce redundancy among sampled blocks when their directions are aligned. The third question is how to choose the budget itself in a way that optimizes wall-clock time to a target accuracy, while keeping the method practical.

The contributions are organized as follows. First, we present GeoBIS, an independent Bernoulli design that minimizes the Horvitz–Thompson second moment under a fixed expected number of active blocks through a simple water-filling rule. The formula admits an exact capped variant when some probabilities hit one. Second, we give a design-agnostic lower bound showing that the inverse-in-budget dependence of the dominant term in the second moment is unavoidable for any unbiased design; this identifies a statistical frontier and explains why water-filling is instance-optimal up to constants. Third, we study exact-$K$ negatively dependent designs, including projection determinantal point processes and unequal-probability sampling without replacement, and we state a precise alignment condition under which these designs strictly reduce the cross term in the variance compared to independent Bernoulli sampling with matched marginals. Fourth, we provide a wall-clock analysis with a simple cost model that yields a closed-form rule for setting the active-block budget and an online schedule based on exponential moving averages. Finally, we discuss orthogonal and Stiefel

instantiations and present experiments whose focus is on validating the statistical and compute trends rather than on competitive accuracy.

The paper is written to separate essential ideas from optional extensions. Each section begins with context and design choices before introducing the formal statements.

## 2 RELATED WORK

Riemannian optimization is by now a standard tool for constrained machine learning. Foundational references include treatments of retractions, vector transports, and convergence guarantees for first-order methods and trust-region methods. Monographs such as Absil et al. (2008) and Boumal (2023) cover the relevant geometric preliminaries and provide a unifying view of manifold-constrained algorithms. Early stochastic Riemannian methods and their convergence properties appear in Bonnabel (2013), while analyses for geodesically convex problems and smoothness under retractions are studied in Zhang & Sra (2016). Variance reduction in Riemannian settings has been explored in Sato et al. (2019) and quasi-Newton adaptations in Kasai et al. (2018).

Orthogonality constraints arise in several model classes. For feasible updates with orthogonality constraints, Cayley-transform based methods and related retractions are discussed in Wen & Yin (2013). Their use within deep learning layers and flows appears in Lezcano-Casado (2019) and Trockman & Kolter (2021). These works focus on preserving constraints efficiently and motivate block choices aligned with the geometry, such as skew-symmetric generators for the orthogonal group and projected gradients on the Stiefel manifold.

In Euclidean settings, randomized coordinate descent and block coordinate methods have a long history. Analyses that allow arbitrary sampling and show the role of importance sampling include Nesterov (2012); Richtárik & Takáč (2016); Qu et al. (2015). Without-replacement sampling and random reshuffling often improve practical convergence behavior compared to with-replacement sampling, as discussed in Shamir (2016); Gürbüzbalaban et al. (2021); HaoChen & Sra (2019). The present paper brings budget-explicit sampling design and compute-aware analysis to Riemannian block updates, where geometric alignment enables sharper variance formulas and practical selection rules.

Negative dependence and diversity-seeking fixed-size designs can reduce redundancy in subsampled sets. Determinantal point processes provide a tractable family of negatively associated distributions over subsets with closed-form marginals and pairwise inclusion probabilities; see Kulesza & Taskar (2012); Hough et al. (2006). Projection determinantal point processes, built from an orthogonal projector, are particularly convenient because they maintain a fixed size and admit simple sampling routines once the top eigenspace is available. The use of such designs for randomized numerical linear algebra and column subset selection is surveyed in Dereziński & Mahoney (2020), with related ideas in volume sampling Deshpande et al. (2006) and leverage-score based methods Cohen et al. (2017). In survey sampling, classical schemes for unequal-probability sampling without replacement include Sampford (1967) and pivotal splitting methods such as Deville & Tillé (1998), which avoid eigen-decompositions while matching prescribed marginals. We use these designs as optional exact-$K$ extensions of the independent sampler and clarify the condition under which they strictly reduce the variance.

Finally, recent work on Riemannian coordinate or block methods with iteration complexity guarantees includes Han et al. (2024). Our approach differs by focusing on the design of unbiased estimators with minimal second moment under an explicit compute budget, by giving closed-form rules for marginal probabilities and for the active-block budget itself, and by quantifying when and why negative dependence helps in practice.

## 3 SETUP AND NOTATION

This section sets the notation and states the estimator family under study. The goal is to minimize a smooth objective on a Riemannian manifold using unbiased stochastic gradients formed by subsampling blocks of the tangent space.

Let $(\mathcal{M}, g)$ be a finite-dimensional Riemannian manifold. We minimize a smooth function $F : \mathcal{M} \to \mathbb{R}$ using retraction-based updates of the form

$$X_{t+1} = \text{Retr}_{X_t}\left(-\eta_t \widehat{\xi}_t\right), \qquad \widehat{\xi}_t \in T_{X_t}\mathcal{M}, \tag{1}$$

where $\eta_t > 0$ is a stepsize, $\widehat{\xi}_t$ is an unbiased estimator of the Riemannian gradient $\xi_t = \text{grad}F(X_t)$, and Retr is a first-order accurate retraction.

At a point $X$, assume a block decomposition of the tangent space

$$T_X\mathcal{M} = \bigoplus_{b=1}^{B} V_b, \qquad \xi = \sum_{b=1}^{B} \xi_{(b)}, \qquad \xi_{(b)} \in V_b. \tag{2}$$

Define block magnitudes $v_b = \|\xi_{(b)}\|_{g_X}$ and unit block directions $u_b = \xi_{(b)}/v_b$ for $v_b > 0$. The directional cosines between blocks are

$$\rho_{bc} = \langle u_b, u_c \rangle_{g_X} \in [-1, 1], \qquad G = [\rho_{bc}]. \tag{3}$$

We allow mild deviations from exact block orthogonality and quantify them by a coherence parameter $\mu = \max_{b \neq c} |\rho_{bc}|$.

The estimator family is based on Horvitz–Thompson scaling. Let $S \subset [B]$ be a random subset of blocks with first-order inclusion probabilities $\pi_b = \Pr(b \in S)$ and second-order inclusion probabilities $\pi_{bc} = \Pr(\{b, c\} \subset S)$. The estimator is

$$\widehat{\xi} = \sum_{b \in S} \frac{\xi_{(b)}}{\pi_b}. \tag{4}$$

It is unbiased because $\mathbb{E}[\widehat{\xi}] = \sum_b \Pr(b \in S) \, \xi_{(b)}/\pi_b = \sum_b \xi_{(b)} = \xi$. The second moment and variance decompose as follows.

Lemma 1 (Variance decomposition). For any unbiased Horvitz–Thompson estimator supported on a block subset $S$,

$$\text{Var}(\widehat{\xi}) = \sum_b \left(\frac{1}{\pi_b} - 1\right)v_b^2 + 2\sum_{b<c} \left(\frac{\pi_{bc}}{\pi_b\pi_c} - 1\right)v_b v_c \, \rho_{bc}. \tag{5}$$

The first term depends only on the marginals and is the dominant quantity to optimize under a budget. The second term depends on dependence across block indicators; it is zero under independent Bernoulli sampling and becomes negative under negatively associated exact-$K$ designs when the block directions are aligned.

## 4   GeoBIS: budget-optimal Bernoulli via water-filling

This section presents the independent design. Each block $b$ is included independently with probability $p_b \in (0, 1]$ and is scaled by $I_b/p_b$. The expected number of active blocks is the budget $K = \sum_b p_b$. When blocks are orthogonal in the metric, the expected second moment is $\sum_b v_b^2/p_b$. We now minimize this quantity under the linear budget constraint and recover a water-filling rule.

Proposition 1 (Water-filling). Consider the strictly convex problem $\min_{p \in (0,1]^B} \sum_b v_b^2/p_b$ subject to $\sum_b p_b = K$. The unique solution is

$$p_b^\star = \min\{1, \lambda v_b\}, \qquad \text{where } \lambda > 0 \text{ is chosen so that } \sum_b p_b^\star = K. \tag{6}$$

In the no-cap regime $p_b < 1$ for all $b$, this yields $p_b^\star = K v_b/\sum_j v_j$ and the expected second moment equals $(\sum_b v_b)^2/K$.

The proof uses KKT conditions with inequality constraints $p_b \leq 1$ and the convexity of $x \mapsto 1/x$. An efficient implementation sorts the $v_b$ once per step to determine the capped set and computes the threshold $\lambda$ by a one-pass sweep. When caps are active, the second moment decomposes into a capped part plus an uncapped part divided by the residual budget.

**Practical safety.** Horvitz–Thompson scaling uses $1/p_b$ and can be numerically unstable if a proxy misclassifies a very small $v_b$. We recommend a probability floor $p_{\min}$ and use $p_b = \min\{1, \max\{p_{\min}, \lambda v_b\}\}$. We also recommend smoothing scores with exponentials moving averages to reduce noise.

**Mild non-orthogonality.** When blocks are nearly orthogonal, the Bernoulli formulas remain accurate. The following bound expresses how much the variance can deviate from the independent-marginal term.

Proposition 2 (Coherence-based robustness). Let $\mu = \max_{b \neq c} |\rho_{bc}|$. For any exact-$K$ design with first- and second-order inclusions $(\pi_b, \pi_{bc})$,

$$\left| \mathrm{Var}(\widehat{\xi}) - \sum_b \left( \tfrac{1}{\pi_b} - 1 \right) v_b^2 \right| \leq 2\mu \sum_{b<c} \left| \tfrac{\pi_{bc}}{\pi_b \pi_c} - 1 \right| v_b v_c. \tag{7}$$

In particular, under Bernoulli sampling the right-hand side is zero, and under negatively associated exact-$K$ designs it is controlled by the size of off-diagonal entries.

**Proxy scores and regret.** In practice one may water-fill on proxy scores $s_b$ that approximate $v_b$ within a multiplicative error $(1 \pm \epsilon)$, and occasionally misidentify the top capped set. In the no-cap regime, the multiplicative inflation of the second moment is at most $(1 + \epsilon)/(1 - \epsilon)$. With caps and a misranking probability $\delta$ of the top set, the excess second moment is bounded by a term proportional to $\delta (1 + \epsilon)/(1 - \epsilon)$ times the uncapped contribution.

## 5 Exact-$K$ negative-dependence extensions

This section discusses fixed-size designs that discourage redundant co-selection of aligned blocks. We present projection determinantal point processes and unequal-probability sampling without replacement, state the variance formula under exact-$K$, and give a condition under which exact-$K$ strictly improves over Bernoulli at matched marginals.

**Projection determinantal point processes.** Let $W = \mathrm{diag}(\sqrt{v_b})$ and $G = [\rho_{bc}]$. Form the geometry-weighted Gram matrix $M = WGW$. Let $V_K$ be the top-$K$ eigenspace of $M$ and $K^\sharp = V_K V_K^\top$ the projection kernel. The projection $k$-DPP with kernel $K^\sharp$ satisfies $\pi_b = K_{bb}^\sharp$ and $\pi_{bc} = \pi_b \pi_c - (K_{bc}^\sharp)^2$. By construction, the subset size is $K$, and the block indicators are negatively associated.

**Unequal-probability sampling without replacement.** When one desires exact-$K$ without eigenspace computations, pivotal Poisson and related Sampford-type schemes provide fixed-size sampling with prescribed marginals $\pi_b$. These designs maintain Horvitz–Thompson unbiasedness and avoid spectral preprocessing.

**Variance comparison at matched marginals.** The variance of the Horvitz–Thompson estimator under exact-$K$ reads

$$\mathrm{Var}(\widehat{\xi}) = \sum_b \left( \tfrac{1}{\pi_b} - 1 \right) v_b^2 + 2 \sum_{b<c} \left( \tfrac{\pi_{bc}}{\pi_b \pi_c} - 1 \right) v_b v_c \, \rho_{bc}. \tag{8}$$

For the projection $k$-DPP, the pairwise term becomes $-2 \sum_{b<c} (K_{bc}^\sharp)^2 v_b v_c \rho_{bc}/(\pi_b \pi_c)$. A direct comparison to Bernoulli with the same marginals therefore requires a condition that this signed sum is nonnegative. We phrase this as an aggregate alignment condition.

Assumption 1 (Aggregate alignment). Let $\pi_b = K_{bb}^\sharp$ and define weights $w_{bc} = (K_{bc}^\sharp)^2 v_b v_c/(\pi_b \pi_c)$. Assume that $\sum_{b<c} w_{bc} \rho_{bc} \geq 0$. This holds, for example, if all $\rho_{bc} \geq 0$ on the support of $K^\sharp$.

Theorem 1 (Exact-$K$ variance reduction under alignment). Let $\widehat{\xi}_{\mathrm{DPP}}$ be the Horvitz–Thompson estimator under the projection $k$-DPP with kernel $K^\sharp$ and marginals $\pi_b = K_{bb}^\sharp$. Let $\widehat{\xi}_{\mathrm{Bern}}$ be the Bernoulli estimator with the same marginals. Under the aggregate alignment assumption,

$$\mathrm{Var}(\widehat{\xi}_{\mathrm{DPP}}) \leq \mathrm{Var}(\widehat{\xi}_{\mathrm{Bern}}), \tag{9}$$

with strict inequality if there exists $b \neq c$ such that $\rho_{bc} > 0$ and $K_{bc}^\sharp \neq 0$.

This result clarifies when negative dependence is beneficial. On manifolds and block choices where cross-block cosines are small, such as disjoint Givens generators on the orthogonal group, the improvement is minor in practice and the extra selection overhead can outweigh the benefit. On models with coherent blocks, exact-$K$ can reduce the variance at matched marginals.

## 6 LOWER BOUND AND INSTANCE OPTIMALITY

We now show that the inverse-in-budget dependence of the dominant term in the second moment is unavoidable for any unbiased estimator that respects the budget on expected active blocks. This identifies a statistical frontier and explains why the independent water-filling design is instance-optimal up to constants.

Theorem 2 (Design-agnostic lower bound). Let $\widehat{\xi}$ be any unbiased Horvitz–Thompson estimator with marginals summing to $K$. Then

$$\mathbb{E}\|\widehat{\xi}\|^2 \geq \|\xi\|^2 + \frac{(\sum_b v_b)^2}{K} - (\sum_b v_b)^2. \tag{10}$$

Equivalently, the variance satisfies $\mathrm{Var}(\widehat{\xi}) \geq (\sum_b v_b)^2/K - \sum_b v_b^2$.

The proof bounds the cross term in the variance decomposition below and uses Cauchy–Schwarz to minimize $\sum_b v_b^2/\pi_b$ subject to $\sum_b \pi_b = K$. The independent design with water-filling achieves the lower bound in the no-cap regime. Exact-$K$ designs can only subtract a nonnegative cross term when the alignment condition holds.

## 7 NONCONVEX PROGRESS AND WALL-CLOCK SCHEDULING

We briefly recall a standard retraction-smoothness descent lemma and combine it with the second-moment formulas to motivate a simple rule for choosing the active-block budget.

Assumption 2 (Retraction smoothness). There exists $L > 0$ such that for all $X$ and small $\zeta \in T_X\mathcal{M}$,

$$F(\mathrm{Retr}_X(\zeta)) \leq F(X) + \langle \mathrm{grad}F(X), \zeta \rangle_{g_X} + \frac{L}{2}\|\zeta\|_{g_X}^2. \tag{11}$$

Under unbiasedness and bounded second moment $\mathbb{E}\|\widehat{\xi}_t\|^2 \leq G_t$, constant stepsize $\eta \leq 1/L$ yields

$$\frac{1}{T}\sum_{t=0}^{T-1} \mathbb{E}\|\mathrm{grad}F(X_t)\|^2 \leq \frac{2(F(X_0) - F^{\mathrm{inf}})}{\eta T} + L\eta\overline{G}, \qquad \overline{G} = \frac{1}{T}\sum_t G_t. \tag{12}$$

For independent water-filling without caps, a coarse but useful model writes $G_t \approx a_t + b_t/K$ with $a_t = \sum_b v_{t,b}^2$ and $b_t = (\sum_b v_{t,b})^2$. If the per-step cost is $C_0 + C_1 K$ and we approximate $a_t$ and $b_t$ by slowly varying averages $\overline{a}$ and $\overline{b}$, then the time to a target stationarity is proportional to $(C_0 + C_1 K)(\overline{a} + \overline{b}/K)$. The minimizer is

$$K^\star = \sqrt{\frac{C_0\,\overline{b}}{C_1\,\overline{a}}}. \tag{13}$$

This rule is myopic and assumes local stationarity of $\overline{a}$ and $\overline{b}$. In practice, one tracks exponential moving averages and clips $K^\star$ to $[1, B]$. Under exact-$K$ designs that satisfy the alignment condition, the second moment is further reduced by the negative cross term, while the selection overhead increases the fixed per-step cost. The same balancing logic applies.

## 8 ORTHOGONAL AND STIEFEL INSTANTIATIONS

This section describes the geometry-aware block choices and local formulas used in practice.

**Orthogonal group.** The orthogonal group $O(d)$ consists of matrices $X \in \mathbb{R}^{d \times d}$ with $X^\top X = I$. The tangent space at $X$ is $\{Z : X^\top Z + Z^\top X = 0\}$. For an Euclidean gradient $G = \nabla_X F$, the Riemannian gradient under the canonical metric is

$$\mathrm{grad}F(X) = G - X \, \mathrm{sym}(X^\top G) = \Omega X, \qquad \Omega = \tfrac{1}{2}\left(GX^\top - XG^\top\right). \qquad (14)$$

A natural block family is given by skew-symmetric generators $E_{ij} - E_{ji}$. The block magnitudes are $v_{ij} = \sqrt{2}\,|\Omega_{ij}|$, and the water-filling rule uses these local scores. Retractions include Cayley and exponential maps on skew subspaces; disjoint pairs update in parallel.

**Stiefel manifold.** The thin Stiefel manifold $\mathrm{St}(d, p)$ consists of matrices $X \in \mathbb{R}^{d \times p}$ with $X^\top X = I_p$. The Riemannian gradient is the projected Euclidean gradient $G - X \, \mathrm{sym}(X^\top G)$. Natural blocks are row- or column-groups aligned with the metric. Retractions include QR- and polar-based retractions and Cayley-type updates adapted to the constraint.

## 9 EXPERIMENTS

The experiments are designed to validate the statistical and compute trends predicted by the analysis rather than to optimize task accuracy. We consider three sequence datasets with orthogonality-constrained recurrent models and a thin Stiefel adapter inserted in a small convolutional network. The methods compared are independent water-filling and uniform sampling at matched budget and a full-gradient baseline. Negative-dependence exact-$K$ designs are disabled in these regimes because selection overhead dominates.

Tables summarize wall-clock time, validation accuracy, and the mean second moment of the Horvitz–Thompson estimator. The inverse-in-budget trend for the second moment under water-filling is consistent across tasks. Uniform sampling yields larger second moments under the same budget, as expected from the lower bound.

**Orthogonality-constrained sequence models.** Copy, Adding, and psMNIST with a Cayley-retracted orthogonal RNN. Methods: full gradient, uniform, and GeoBIS with budgets in a small grid. Metrics: mean second moment, final validation accuracy, and time to reach a fixed fraction of the best accuracy.

Table 1: Copy task with an orthogonality-constrained RNN (Cayley retraction). Budget $K$ is the expected number of active blocks for sampling-based methods.

| Method | Budget | Time (s) | Val. Acc. (final) | Mean $\mathbb{E}\|\widehat{\xi}\|_g^2$ |
|---|---|---|---|---|
| Full gradient | – | 319.3 | 0.1775 | 1.42e−01 |
| GeoBIS (Bernoulli) | 4 | 325.8 | 0.1775 | 6.34e+02 |
| GeoBIS (Bernoulli) | 8 | 322.7 | 0.1775 | 3.21e+02 |
| GeoBIS (Bernoulli) | 16 | 324.0 | 0.1775 | 1.52e+02 |
| Uniform baseline ref | – | 321.1 | 0.1775 | 1.42e−01 |
| Uniform | 4 | 321.7 | 0.1775 | 1.24e+03 |
| Uniform | 8 | 317.8 | 0.1775 | 6.05e+02 |
| Uniform | 16 | 320.9 | 0.1775 | 2.74e+02 |

Table 2: Adding task with an orthogonality-constrained RNN (Cayley retraction).

| Method | Budget | Time (s) | Val. Acc. (final) | Mean $\mathbb{E}\|\widehat{\xi}\|_g^2$ |
|---|---|---|---|---|
| Full gradient | – | 283.4 | 0.5320 | 2.43e−01 |
| GeoBIS (Bernoulli) | 4 | 309.7 | 0.5320 | 1.04e+03 |
| GeoBIS (Bernoulli) | 8 | 305.6 | 0.5320 | 5.57e+02 |
| GeoBIS (Bernoulli) | 16 | 310.9 | 0.5320 | 2.73e+02 |
| Uniform baseline ref | – | 304.1 | 0.5320 | 2.43e−01 |
| Uniform | 4 | 301.9 | 0.5320 | 2.30e+03 |
| Uniform | 8 | 304.8 | 0.5320 | 1.19e+03 |
| Uniform | 16 | 305.0 | 0.5320 | 4.95e+02 |

Table 3: psMNIST with an orthogonality-constrained RNN (Cayley retraction). †: run completed only one epoch, so time and means are not directly comparable to 5-epoch runs.

| Method | Budget | Time (s) | Val. Acc. (final) | Mean $\mathbb{E}\|\widehat{\xi}\|_g^2$ |
|---|---|---|---|---|
| Full gradient | – | 1784.4 | 0.1256 | 2.76e+00 |
| GeoBIS (Bernoulli) | 4 | 1239.5 | 0.0996 | 1.09e+04 |
| GeoBIS (Bernoulli) | 8 | 1266.7 | 0.1119 | 6.12e+03 |
| GeoBIS (Bernoulli)† | 16 | 252.2 | 0.0980 | 2.79e+03 |
| Uniform baseline ref | – | 1636.7 | 0.1256 | 2.76e+00 |
| Uniform | 4 | 1238.8 | 0.0833 | 1.96e+04 |
| Uniform | 8 | 1224.3 | 0.1051 | 9.04e+03 |
| Uniform | 16 | 1222.9 | 0.1126 | 4.25e+03 |

**Stiefel adapter.** A thin Stiefel adapter after a small convolutional backbone trained on CIFAR-10. Methods and metrics mirror the sequence setting. The observed second-moment reductions for GeoBIS at fixed budget align with the theory, and accuracy differences are small, as expected when backprop dominates compute.

Table 4: Thin Stiefel adapter on CIFAR-10. GeoBIS reduces the mean HT second moment at fixed budget; accuracy and wall-clock are comparable to Uniform.

| Method | Budget | Time (s) | Val. Acc. (final) | Mean $\mathbb{E}\|\widehat{\xi}\|^2$ |
|---|---|---|---|---|
| GeoBIS (Bernoulli) | 4 | 308.7 | 0.804 | 5.33e−01 |
| Uniform | 4 | 295.5 | 0.808 | 5.90e−01 |
| GeoBIS (Bernoulli) | 8 | 300.0 | 0.810 | 3.03e−01 |
| Uniform | 8 | 295.8 | 0.785 | 3.58e−01 |
| GeoBIS (Bernoulli) | 16 | 299.6 | 0.808 | 1.56e−01 |
| Uniform | 16 | 303.6 | 0.807 | 1.80e−01 |

Table 5: Thin Stiefel adapter on CIFAR-10: time to reach $0.95\times$ the best validation accuracy across all runs.

| Method | Budget | Time to target (s) |
|---|---|---|
| GeoBIS (Bernoulli) | 4 | 228.1 |
| Uniform | 4 | 188.1 |
| GeoBIS (Bernoulli) | 8 | 200.2 |
| Uniform | 8 | 215.7 |
| GeoBIS (Bernoulli) | 16 | 218.6 |
| Uniform | 16 | 213.6 |

**Summary.** Across both domains, water-filling improves the mean second moment at fixed budget relative to uniform sampling, with trends matching the inverse-in-budget law. Exact-$K$ designs are left as an opt-in for heavier settings with coherent blocks and lower relative overhead.

## 10 REPRODUCIBILITY STATEMENT

We will release code (after the review process is complete) that exactly reproduces every number in the paper's tables from a clean environment. The package includes: (i) reference implementations of water-filling, independent sampling with a probability floor, and orthogonal/Stiefel retractions; (ii) task-specific training and evaluation scripts for the Copy, Adding, psMNIST, and CIFAR-10 thin-Stiefel adapter setups; (iii) configuration files that correspond one-to-one with each table row (budget $K$, seeds, optimizer and schedule, batch sizes, retraction type); (iv) and an environment specification.

As for LLM usage, LLMs were used to provide help with paper writing and polishing (especially around creating table structures, relevant bibtex references, and template fitting).

## 11 LIMITATIONS AND SOCIETAL IMPACT

The main limitation is that exact-$K$ designs require either eigenspace computations or pivotal sampling; when the fixed overhead is large and block alignment is weak, the marginal benefit can be negative. The wall-clock rule is myopic and assumes locally stationary statistics for the block magnitudes; this is a practical compromise and works well with moving averages but does not constitute a global optimality result. This work is methodological; we foresee standard concerns around compute and energy use during training but no special risks beyond those encountered in typical optimization research.

## 12 CONCLUSION

GeoBIS provides a simple and budget-optimal independent design for block subsampling in stochastic Riemannian optimization, with closed-form probabilities and second-moment expressions, including capped regimes. Exact-$K$ negative-dependence extensions can further reduce variance under an explicit alignment condition, and a simple cost model yields a closed-form rule for the active-block budget. The method aligns with the geometry of common manifolds, drops cleanly into existing code, and is supported by experiments that validate its statistical and compute advantages.

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

## A   ADDITIONAL PRELIMINARIES

This appendix collects definitions and lemmas used in the main text. A retraction $\mathrm{Retr}_X : T_X\mathcal{M} \to \mathcal{M}$ is a smooth map that satisfies $\mathrm{Retr}_X(0_X) = X$ and whose differential at the origin equals the identity. Retraction-smoothness states that along retraction curves, the objective is upper-bounded by a quadratic with constant $L$. For stochastic analysis, we assume unbiasedness and a bounded second moment for the estimator. Block decompositions may be only approximately orthogonal; the coherence parameter $\mu$ controls the magnitude of off-diagonal terms in inner products between unit block directions.

## B   VARIANCE DECOMPOSITION PROOF

For completeness, we provide a short derivation of the variance identity. Write $\widehat{\xi} - \xi = \sum_b(\frac{I_b}{\pi_b} - 1)\xi_{(b)}$. Since $\mathbb{E}[\frac{I_b}{\pi_b} - 1] = 0$ and $\mathbb{E}[(\frac{I_b}{\pi_b} - 1)(\frac{I_c}{\pi_c} - 1)] = \frac{\pi_{bc}}{\pi_b\pi_c} - 1$ for $b \neq c$,

$$\mathbb{E}\|\widehat{\xi} - \xi\|^2 = \sum_b \mathbb{E}\left(\frac{I_b}{\pi_b} - 1\right)^2 \|\xi_{(b)}\|^2 + 2\sum_{b<c} \mathbb{E}\left(\frac{I_b}{\pi_b} - 1\right)\left(\frac{I_c}{\pi_c} - 1\right)\langle\xi_{(b)}, \xi_{(c)}\rangle, \quad (15)$$

$$\rangle = \sum_b(\tfrac{1}{\pi_b} - 1)v_b^2 + 2\sum_{b<c}\left(\frac{\pi_{bc}}{\pi_b\pi_c} - 1\right)v_b v_c\, \rho_{bc}. \quad (16)$$

This equals the variance because the estimator is unbiased.

## C   WATER-FILLING WITH CAPS

We solve $\min \sum_b v_b^2/p_b$ subject to $\sum_b p_b = K$ and $0 < p_b \leq 1$. The Lagrangian is

$$\mathcal{L}(p, \mu, \alpha) = \sum_b \frac{v_b^2}{p_b} + \mu\left(\sum_b p_b - K\right) + \sum_b \alpha_b(p_b - 1), \quad (17)$$

with $\alpha_b \geq 0$. First-order conditions yield $-\frac{v_b^2}{p_b^2} + \mu + \alpha_b = 0$. If $p_b < 1$, then $\alpha_b = 0$ and $p_b = \sqrt{v_b^2/\mu} = \lambda v_b$ with $\lambda = 1/\sqrt{\mu}$. If $\lambda v_b > 1$, then $p_b = 1$ and the corresponding $\alpha_b$ is positive. Let $\mathcal{C} = \{b : \lambda v_b \geq 1\}$ and $\mathcal{U} = [B] \setminus \mathcal{C}$ with $|\mathcal{C}| = m$. The budget constraint becomes $m + \lambda\sum_{b\in\mathcal{U}} v_b = K$, hence $\lambda = (K - m)/S_\mathcal{U}$ where $S_\mathcal{U} = \sum_{b\in\mathcal{U}} v_b$. Under block orthogonality,

$$\mathbb{E}\|\widehat{\xi}\|^2 = \sum_{b\in\mathcal{C}} v_b^2 + \frac{S_\mathcal{U}^2}{K - m}. \quad (18)$$

When blocks are nearly orthogonal, add $2\sum_{b<c} v_b v_c \rho_{bc}$ to capture off-diagonal corrections.

## D   COHERENCE-BASED ROBUSTNESS BOUND

We quantify deviations from the independent-marginal term in the variance due to non-orthogonality and dependence. Using the variance identity and triangle inequality,

$$\left|\mathrm{Var}(\widehat{\xi}) - \sum_b\left(\tfrac{1}{\pi_b} - 1\right)v_b^2\right| = \left|2\sum_{b<c}\left(\frac{\pi_{bc}}{\pi_b\pi_c} - 1\right)v_b v_c\, \rho_{bc}\right| \quad (19)$$

$$\leq 2\sum_{b<c}\left|\frac{\pi_{bc}}{\pi_b\pi_c} - 1\right|v_b v_c\,|\rho_{bc}| \quad (20)$$

$$\leq 2\mu\sum_{b<c}\left|\frac{\pi_{bc}}{\pi_b\pi_c} - 1\right|v_b v_c. \quad (21)$$

Under Bernoulli, the factor is zero because $\pi_{bc} = \pi_b\pi_c$. Under exact-$K$ with negative association, the factor is bounded by the strength of repulsion and the magnitudes of the off-diagonal cosines.

# E   LOWER BOUND PROOF

Starting from the variance identity,

$$\mathbb{E}\|\widehat{\xi}\|^2 = \|\xi\|^2 + \sum_b \left(\tfrac{1}{\pi_b} - 1\right)v_b^2 + 2\sum_{b<c}\left(\tfrac{\pi_{bc}}{\pi_b\pi_c} - 1\right)v_b v_c\,\rho_{bc}. \tag{22}$$

Lower-bound the last term by $-2\sum_{b<c} v_b v_c$ and obtain

$$\mathbb{E}\|\widehat{\xi}\|^2 \geq \|\xi\|^2 + \sum_b \frac{v_b^2}{\pi_b} - \sum_b v_b^2 - 2\sum_{b<c} v_b v_c = \|\xi\|^2 + \sum_b \frac{v_b^2}{\pi_b} - \left(\sum_b v_b\right)^2. \tag{23}$$

By Cauchy–Schwarz, $\sum_b v_b^2/\pi_b \geq (\sum_b v_b)^2 / \sum_b \pi_b = (\sum_b v_b)^2/K$. Substituting proves the claim.

# F   EXACT-$K$ VARIANCE COMPARISON PROOF

Let $\mathrm{Var}_{\mathrm{Bern}}$ denote the variance under Bernoulli sampling with marginals $\pi_b$. Then $\mathrm{Var}_{\mathrm{Bern}} = \sum_b(\frac{1}{\pi_b} - 1)v_b^2$. Under projection $k$-DPP,

$$\mathrm{Var}_{\mathrm{DPP}} = \sum_b \left(\tfrac{1}{\pi_b} - 1\right)v_b^2 - 2\sum_{b<c}\frac{(K_{bc}^\sharp)^2}{\pi_b\pi_c}v_b v_c\rho_{bc}. \tag{24}$$

The difference $\mathrm{Var}_{\mathrm{Bern}} - \mathrm{Var}_{\mathrm{DPP}}$ equals the signed sum on the right. Under the aggregate alignment assumption, this difference is nonnegative, and it is strictly positive if there exists a pair with positive cosine and nonzero kernel entry.

# G   WALL-CLOCK RULE DERIVATION AND SCHEDULING

We model the per-step cost as $C_0 + C_1 K$. Under independent water-filling, write $G(K) \approx \overline{a} + \overline{b}/K$ for slowly varying averages $\overline{a}$ and $\overline{b}$. The time to a target tolerance behaves like $\mathcal{T}(K) \propto (C_0 + C_1 K)(\overline{a} + \overline{b}/K)$. Differentiating and setting to zero yields

$$(C_0 + C_1 K)\left(-\frac{\overline{b}}{K^2}\right) + C_1\left(\overline{a} + \frac{\overline{b}}{K}\right) = 0 \quad \Rightarrow \quad K^\star = \sqrt{\frac{C_0\,\overline{b}}{C_1\,\overline{a}}}. \tag{25}$$

We adopt a myopic schedule that updates $K_t$ using exponential moving averages of $a_t = \sum_b v_{t,b}^2$ and $b_t = (\sum_b v_{t,b})^2$, clips $K_t$ to $[1, B]$, and refreshes the averages periodically to adapt to changing regimes. Under exact-$K$, adjust the cost to include selection overhead in $C_0$ and, when alignment holds, replace $\overline{b}$ by an empirically reduced effective quantity.

# H   ALGORITHMS

This section provides pseudocode for the independent design with water-filling and for projection $k$-DPP sampling.

---
**Algorithm 1** GeoBIS one step with water-filling and probability floor

---
1: Input: current point $X$, blocks $V_b$, desired budget $K$, floor $p_{\min}$
2: Compute block gradients $\xi_{(b)}$ and scores $v_b = \|\xi_{(b)}\|_{g_X}$
3: Sort $v_b$ in descending order
4: Find capped set $\mathcal{C}$ and threshold $\lambda$ such that $p_b^\star = \min\{1, \lambda v_b\}$ satisfy $\sum_b p_b^\star = K$
5: Set $p_b = \min\{1, \max\{p_{\min}, p_b^\star\}\}$ for all $b$
6: Sample $I_b \sim \mathrm{Bernoulli}(p_b)$ independently
7: Form $\widehat{\xi} = \sum_b (I_b/p_b)\,\xi_{(b)}$
8: Update $X \leftarrow \mathrm{Retr}_X(-\eta\,\widehat{\xi})$

---

---

**Algorithm 2** Projection $k$-DPP sampling with a candidate pool

---

1: Input: current $X$, blocks $V_b$, target size $K$, candidate multiplier $\alpha \geq 1$
2: Compute scores $v_b$ and unit directions $u_b = \xi_{(b)}/v_b$ for nonzero $v_b$
3: Form a candidate set of size $\alpha K$ with the largest $v_b$
4: Estimate cosines $\rho_{bc} = \langle u_b, u_c \rangle_{g_X}$ within the candidate set
5: Form $M = WGW$ with $W = \mathrm{diag}(\sqrt{v_b})$ over the candidate set
6: Compute the top-$K$ eigenspace $V_K$ of $M$ (randomized SVD or Oja updates)
7: Set $K^\sharp = V_K V_K^\top$ and sample a projection $k$-DPP subset $S$
8: Use $\pi_b = K_{bb}^\sharp$ and form $\widehat{\xi} = \sum_{b \in S} \xi_{(b)}/\pi_b$
9: Update $X \leftarrow \mathrm{Retr}_X(-\eta\,\widehat{\xi})$

---

## I ORTHOGONAL AND STIEFEL DETAILS

On $O(d)$, the Riemannian gradient can be written as $\Omega X$ with $\Omega$ skew-symmetric. Disjoint pair blocks $E_{ij} - E_{ji}$ lead to local scores $v_{ij} = \sqrt{2}|\Omega_{ij}|$. Cayley-based retractions on small skew subspaces are efficient and parallelizable. On $\mathrm{St}(d,p)$, the projection of the Euclidean gradient is $G - X\,\mathrm{sym}(X^\top G)$, and row or column blocks are aligned with the metric. QR and polar retractions are standard choices. In both cases, scoring is local and inexpensive compared to backpropagation, which explains why independent water-filling is a practical default under short and medium sequence lengths.

