# OpenReview forum: "GeoBIS: Budget-Optimal Block Importance Sampling for Stochastic Riemannian Optimization"
_ICLR.cc/2026/Conference — ICLR 2026 Conference Withdrawn Submission_

### Official Review · Reviewer_nZMq · 2025-10-30

**Soundness:** 2
**Presentation:** 1
**Contribution:** 2
**Rating:** 2
**Confidence:** 4

**Summary:**

In this paper, the authors studied optimization problems over Riemannian manifolds. Based on the Riemannian gradient descent, the authors aimed to replace the Riemannian gradient at each iteration by a block-estimate gradient to reduce the computational cost (for example in high dimensions). Note that block estimate is possible since the tangent space is flat and admits (orthogonal) decompositions. The main contribution is to design strategies to (stochastically) selecting blocks at each iteration, rather than simply using uniformly randomly selected blocks. The authors considered the Horvitz-Thompson estimator (basically the weighted sum estimator in order to ensure unbiasedness) within two block selection strategies: Bernoulli sampling and exact-K. The main constraint of selection blocks is that the expected selected number of blocks is fixed (aka. the budget, and is equals to K given by a user). In the Bernoulli case, they proposed selecting a set of blocks that satisfies the budget constraint while having the smallest variance -- in which they called optimal and denoted by GeoBIS.

Theoretically, they showed:

- The variance in GeoBIS scales as $O(1/K)$  where $K$ is the budget, meaning that if the budget is small, the variance has to be large.
- The variance of any Horvitz-Thompson estimator is at least $O(1/K)$, meaning that GeoBIS is optimal **in terms of** $K$.
- Under the aggregate alignment assumption on the block structure of tangent spaces, the variance of K-exact strategy is smaller than that of GeoBIS.

Finally, some experiments are conducted to study statistical properties of the proposed estimators.

**Strengths:**

- The GeoBIS is optimal in the context of Bernoulli sampling while admiting an implementable form.

**Weaknesses:**

- The writing is bad and requires major revision. Except for "Related work" section, the paper does not contain any other references, either in the introduction, method development, or experiments. Therefore, the paper does not meet the standard scientific bar yet.

- The authors argued that block gradients help in high dimensional settings and this is the main motivation. In the experiments, however, it does not help improve the performance of full gradient and sometimes it is even slower. The only thing the experiments can demonstrate is that the variance of GeoBIS is smaller than the standard uniform strategy. Also, the valiation accuracies in these experiments are too low and hence not convincing.

- Although the paper tried to tackle the computation of the Riemannian gradient within the framework of Riemannian gradient descent, I think another major bottleneck of Riemannian gradient descent lies in the computation of the retraction/exponential map that remains unaddressed.

**Questions:**

- Is $p_b$ in Section 4 the same as $\pi_b$ in Section 3? If so, please make them consistent.
- In line 149, should the variance be $\sum_{b}{v_b^2/p_b} - \sum_{b}{v_b^2}$ instead? The second term is constant nevertheless.
- In equation (7), is the RHS being zero under Beroulli sampling because $\pi_{bc} = \pi_b \pi_c$? If so, it's worth mentioning.
- What is the full name for k-DPP?
- In the experiments, why does higher budget sometimes lead to faster run time? I would imagine the vice versa.

---

### Official Review · Reviewer_u7p1 · 2025-10-31

**Soundness:** 3
**Presentation:** 3
**Contribution:** 2
**Rating:** 4
**Confidence:** 3

**Summary:**

The paper considers the challenge of designing variance-minimizing unbiased block-importance sampling gradient estimators on manifolds.  Such manifold constraints are important for a range of machine learning applications, e.g., in context of orthogonality constraints. For the case of independent sampling, a greedy water-filling design is established to be optimal in the case of independent blocks.  Robustness of the result in the nearly orthogonal setting is established as well.  Next, a family of estimators based on negative dependence, in particular based on projection-DPPs is proposed which is guaranteed to reduce the variance under a particular alignment assumption.  The upper bounds are complemented by a design agnostic lower bound. The estimators are demonstrated on experiments with orthogonality constrained RNNs, on (relatively small for modern standards) psMNIST and CIFAR10 data sets. They demonstrate some advantages of the independent water-filling design over uniform sampling.  The exact-k design is not tested.

**Strengths:**

- stochastic optimization over manifolds is an important problem
- the constructed gradient estimators are simple to implement
- upper and lower bounds on variance are provided

**Weaknesses:**

- The case of negative dependence / exact-k sampling is not empirically evaluated; no compelling case for its utility is made.
- Independent sampling only controls the budget in expectation, which can be problematic when utilizing a fixed number of cores for parallelization

**Questions:**

- Independent sampling has the disadvantage that the budget is only met in expectation.  Doesn’t this cause challenges for parallel computation on hardware? It seems the comparison to uniform sampling is not quite fair for this reason?
- Section 5 remarks that exact-k sampling via pivotal Poisson schemes is possible – this should cause minimal overhead? Why wasn’t this experimentally demonstrated?

---

### Official Review · Reviewer_3jYY · 2025-11-01

**Soundness:** 2
**Presentation:** 2
**Contribution:** 1
**Rating:** 2
**Confidence:** 3

**Summary:**

This paper considers  a budget-optimal block sampling method for stochastic Riemannian optimization, where updates occur on manifolds with structured tangent spaces (e.g., orthogonal or Stiefel constraints).

The authors design an independent Bernoulli sampling rule—derived via a water-filling solution—that minimizes the second moment of the stochastic gradient estimator under a fixed expected budget of active blocks.

Finally, they validate their approach on orthogonality-constrained RNNs.

**Strengths:**

1.The proposed rule is simple to implement and empirically robust across several Riemannian optimization tasks.

2. The authors include detailed algorithmic pseudocode, and plan to release code.

**Weaknesses:**

1. Many symbols in this paper are undefined, such as (V_b, B) on Line 118, (g_X) on Line 122, and (u_c) on Line 123.

2. Several key concepts, including the Horvitz–Thompson estimator, the water-filling probability rule, exact-K negatively dependent designs, projection determinantal point processes, and the wall-clock model, are not adequately explained. As a result, readers without prior background may find it difficult to grasp the paper’s contributions.

3. The paper does not present any significant theoretical results.

4. The authors did not provide the experimental code, raising concerns about the reliability of the results.

**Questions:**

NA

---

### Official Review · Reviewer_BQdm · 2025-11-04

**Soundness:** 2
**Presentation:** 2
**Contribution:** 2
**Rating:** 4
**Confidence:** 3

**Summary:**

The authors investigate optimization under manifold constraints, subject to a budget on the number tangent space blocks (e.g. bases c.f. eq. 2) that are considered during each gradient step.

Their approach, GeoBIS, is based upon the Horvitz–Thompson estimator (eq. 4), and aims to reduce the variance of the estimated gradient (eq. 5) by minimizing the importance-weighted expected second moment (ESM) of the block magnitudes of the tangent space, subject to total block budget K (eq. 6).

Results on several tasks (Tables 1-5) demonstrate that for a given budget on number of blocks (4,8,16), their approach is able to reliably reduce the expected second moment (ESM) of the estimated gradient relative to random sampling, with both on par and mixed results wrt validation error, depending on the task.

Notably, both the ESM and validation accuracy are significantly improved when full gradients are utilized, and the use of full gradients does not require significantly more time, with the exception of one result (the results on psMNIST with an orthogonality-constrained RNN (Cayley retraction) reported in Table 3, where the block methods are about 50% more efficient).

While their experiments focus on their independent Bernoulli design, extensions that take into account block correlation between tangent space blocks are considered in section 5, and a lower bound on gradient variance, which is satisfied by their independent Bernoulli design, is presented in section 6. A simple rule for selecting the block budget K is described in section 7, but is not utilized in the presented experiments.

**Strengths:**

- The optimization of models subject to manifold constraints is interesting and general problem.
- The presented formulation and theoretical results appear sound.
- Results indicate that the approach is reducing the expected second moment of the estimated gradient relative to random block sampling for a given budget K on blocks, as desired.

**Weaknesses:**

- Their results seem to indicate that using full gradients generally takes about the same amount of time, performs much better, and has lower ESM than their approach. If this is the case, why utilize GeoBIS? The ultimate significance of the work needs to b
e further clarified.
- As noted by the authors in section 4 (Practical Safety), the importance weighting in the Horvitz–Thompson estimator makes it inherently less stable, and moving averages and limiters are required. The approach is unbiased (before safety measures are applied), but wouldn't a biased "top-k" (or other) variant of the approach be more effective? Such results, if in fact inferior, could further justify sampling, the approach taken, and the aim to reduce the variance that results. The abstract's claim that GeoBIS "a practical default" for the problem has not been adequately established.
- As emphasized, the paper does not focus on competitive accuracy, but including results without the manifold constraints on one or more tasks would also help to establish the significance of the work.

**Questions:**

See previous section.

---

### Note · Authors · 2025-11-20

**Comment:**

Dear reviewers, after careful consideration, we have decided to withdraw the our submission.
We sincerely thank you for the time you already invested in reviewing this work.

**Withdrawal Confirmation:**

I have read and agree with the venue's withdrawal policy on behalf of myself and my co-authors.